# Antibacterial activity of medicinal plants in Indonesia on *Streptococcus pneumoniae*

**Wisnu Tafroji** [1,2]*, **Nur Ita Margyaningsih**[1], **Miftahuddin Majid Khoeri**[1,3], **Wisiva Tofriska Paramaiswari** [1], **Yayah Winarti**[1], **Korrie Salsabila**[1,2], **Hanifah Fajri Maharani Putri**[1], **Nurjati Chairani Siregar**[1,4], **Amin Soebandrio** [1,4], **Dodi Safari**[1]

**1** Eijkman Institute for Molecular Biology, Jakarta, Indonesia, **2** Master's Programme in Biomedical Sciences, Faculty of Medicine, Universitas Indonesia, Jakarta, Indonesia, **3** Doctoral Program in Biomedical Science, Faculty of Medicine, Universitas Indonesia, Jakarta, Indonesia, **4** Faculty of Medicine, Universitas Indonesia, Jakarta, Indonesia

* wisnutafroji@gmail.com

**Data Availability Statement:** The data underlying the results presented in the study are available at Figshare that can be accessed with following DOI: 10.6084/m9.figshare.20364249 (https://figshare.

## Abstract

*Streptococcus pneumoniae* is a human pathogenic bacterium able to cause invasive pneumococcal diseases. Some studies have reported medicinal plants having antibacterial activity against pathogenic bacteria. However, antibacterial studies of medicinal plants against *S. pneumoniae* remains limited. Therefore, this study aims to describe the antibacterial activity of medicinal plants in Indonesia against *S. pneumoniae*. Medicinal plants were extracted by maceration with n-hexane, ethanol, ethyl acetate and water. Antibacterial activity was defined by inhibition zone and minimum inhibitory concentration (MIC). Bactericidal activity was measured by culture and time-killing measurement. Methods used to describe the mechanism of action of the strongest extract were done by absorbance at 595 nm, broth culture combined with 1% crystal violet, qRT-PCR targeting *lytA*, *peZT* and *peZA*, and transmission electron microscope to measure bacterial lysis, antibiofilm, *LytA* and *peZAT* gene expression, and ultrastructure changes respectively. Among 13 medicinal plants, *L. inermis* Linn. ethyl acetate extract showed the strongest antibacterial activity against *S. pneumoniae* with an MIC value of 0,16 mg/ml. Bactericidal activity was observed at 0,16 mg/ml for 1 hour incubation. *Lawsonia inermis* extract showed some mechanism of actions including bacterial lysis, antibiofilm, and ultrastructure changes such as cell wall disruption, decreasing cell membrane integrity and morphological disorder. Increasing of *lytA* and decreasing of *peZA* and *peZT* expression were also observed after incubation with the extract. In addition, liquid chromatography mass spectrophotometer showed phenolic compounds as the commonest compound in *L. inermis* ethyl acetate extract. This study describes the strong antibacterial activity of *L. inermis* with various mechanism of action including ultrastructure changes.

## Introduction

Bacterial infections have been reported as one of the diseases causing deaths in children under 5 years of age. Pneumonia, sepsis, meningitis and diarrhea were reported as bacterial infection that have caused 2.8 million deaths in neonates [1, 2]. In addition, lower respiratory infections

com/articles/dataset/Minimal_Data_Set_PONE-D-22-12448_xlsx/20364249).

**Funding:** WT received funding from National Geographic Society through The Young Explorer Grants; Grant no. HJ-081ER-17 (https://www.nationalgeographic.org/) and Indonesia Toray Science and Technology Research Grant 2018 (https://itsf.or.id/winners). DS received funding from National Research and Innovation Agency Republic of Indonesia (https://international.brin.go.id/) The funders had no role in study design, data collection and analysis, decision to publish, or preparation of the manuscript.

**Competing interests:** The authors have declared that no competing interests exist.

were reported as the most common bacterial infection disease in developing countries. The mortality and morbidity of bacterial infections can fortunately be reduced by antibiotic therapy [3]. However, the use of antibiotic may lead to the emergence of antibiotic resistant strains as a result of bacterial adaptation against antibiotic pressure due to inappropriate dose, pharmacokinetic, and pharmacodynamic of the antibiotic during antibiotic regiment [4]. This adaptation has resulted in some drug resistant strains such as Methicillin Resistant *Staphylococcus aureus*, which was observed in 1962 after the introduction of penicillin in 1942. Tetracycline resistant strain of *Shigella* was also reported 9 years after tetracycline introduction (1950) [5]. Moreover, *Streptococcus pneumoniae* has also developed resistance to some antibiotics such as penicillin (in 1960), erythromycin (1968), and levofloxacin (1996). Furthermore, multi drug resistant *Streptococcus pneumoniae* was discovered in 1998 after penicillin, erythromycin, cephalosporin, and levofloxacin introduction [6, 7]. Drug resistant strains have also been reported to cause bacterial diseases. In 2013, the Centers for Diseases Control and Prevention (CDC) reported that 2 million cases of bacterial infections were caused by drug resistant strains causing 23000 deaths. In 2007, there were 400000 people infected by multi-drug resistant strains causing 25000 deaths in Europe [8]. Many drug resistant strains of *Streptococcus pneumoniae* were reported in Indonesia such as tetracycline resistant of *S. pneumoniae* that was reported in 2001, other commonly reported drug resistant strains of *S. pneumoniae* include those that are resistant to both cotrimoxazole and tetracycline [9–11]. These reports showed the necessity of exploring other antimicrobial candidates for potential compounds as antibiotic.

Medicinal plants have been reported to have antibacterial activity against some human pathogenic bacteria such as *Staphylococcus aureus*, *Pseudomonas aeruginosa*, *Escherichia coli*, and *Enterococcus sp.* [12–14]. Several developing countries use medicinal plants as a first medicinal response in certain diseases including bacterial infection diseases [15]. There were reports showing that medicinal plants have antibacterial activity against some pathogenic bacteria. *Alchornea cordifolia* showed antibacterial activity against *Escherichia coli*, *P. aeruginosa* and *S. aureus* [16]. *Lippia adoensis*, *Polysphaeria aethiopica*, *Cucumis pustulatus*, *Discopodium penninervium*, *Rumex abyssinicus*, *Cirsium englerianum*, and *Euphorbia depauperate* were reported to have antibacterial activity against *S. aureus*, *E. coli*, *Klebsiella pneumoniae*, and *Streptococcus pyogenes* [17]. *Lawsonia inermis* Linn. was also reported of having antibacterial activity against *S. aureus* and *P. aeruginosa* [18–20]. However, there is limited study in the antibacterial activity of Indonesian medicinal plants against *S. pneumoniae*. Therefore, this study is aimed to investigate the antibacterial activity of medicinal plants in Indonesia against *S. pneumoniae* including its drug resistant strain and to determine the mechanism of action of the most potential antimicrobial candidate among medicinal plants investigated.

## Materials and methods

### Study materials and reagents

The study was conducted according to local guidelines and regulation. The permission to collect the plants was obtained from local authorities and local citizen owning the medicinal plants from each site. The medicinal plants were collected from Gunungkidul and Yogyakarta, D.I. Yogyakarta; Wakatobi, Southeast Sulawesi; Flores, East-Nusa Tenggara; and Manokwari, West Papua. Medicinal plants were collected according to local citizen information about use of plant as alternative medication. All collected medicinal plants were not included in the list of vulnerable or endangered species. The plants were identified according to their morphology and structure of plants as well as assistance of local citizens. Plants and parts of plants collected for this study are listed in Table 1. All the specimens were air-dried under sunlight until further

**Table 1. Medicinal plants used by local citizen in Manokwari, Flores, Wakatobi, Gunungkidul and Yogyakarta.**

| No | Plant species name | Local Name | Collected part | Origin of area | Province | Plant extracts | | | |
|---|---|---|---|---|---|---|---|---|---|
| | | | | | | N-hexane | Ethyl acetate | Ethanol 96% | aqueous |
| 1 | *Drimys piperita* | Akway | Bark | Manokwari | West Papua | Yes | Yes | Yes | Yes |
| 2 | *Myrmecodia pendans* | Sarang semut | Bark | Manokwari | West Papua | Yes | Yes | Yes | Yes |
| 3 | *Biophytum petersianum* Klotzsch | Rumput kebar | Leaf | Manokwari | West Papua | Yes | Yes | Yes | Yes |
| 4 | *Hibiscus sabdarifa* | Rosella | Flower | Yogyakarta | D.I. Yogyakarta | Yes* | Yes | Yes | Yes |
| 5 | *Andrographis paniculata* | Sambiloto | Leaf | Gunungkidul | D.I. Yogyakarta | Yes | Yes | Yes | Yes* |
| 6 | *Piper ornatum* | Sirih Merah | Leaf | Yogyakarta | D.I. Yogyakarta | Yes | Yes | Yes | Yes |
| 7 | *Piper retrofractum* | Cabai Jawa | Fruit | Yogyakarta | D.I. Yogyakarta | Yes | Yes | Yes | Yes |
| 8 | *Lantana camara* | Karuhi-ruhi | Leaf | Wakatobi | Southeast Sulawesi | Yes | Yes | Yes | Yes |
| 9 | *Lawsonia inermis* | Pati-Rangga | Leaf | Wakatobi | Southeast Sulawesi | Yes | Yes | Yes | Yes |
| 10 | *Passiflora foetida* L. | Bambakuru | Leaf | Wakatobi | Southeast Sulawesi | Yes | Yes | Yes* | Yes |
| 11 | *Corypha utan* | Gebang | Leaf | Flores | East Nusa Tenggara | **N/A** | Yes | Yes | **N/A** |
| 12 | **undefined** | Sinsus | Leaf | Flores | East Nusa Tenggara | **N/A** | Yes | Yes | **N/A** |
| 13 | *Vernonia amygdalina* | Bitter leaves | Leaf | Flores | East Nusa Tenggara | **N/A** | Yes | Yes | **N/A** |

*Not processed for antibacterial activity testing

N/A: not done for extraction

processing. The solvents used to extract the specimen were n-Hexane, Ethyl acetate, Ethanol 96% and distillated water.

## Bacterial strain and culture

*Streptococcus pneumoniae* ATCC 49619 and the multidrug resistant *S. pneumoniae* (MDRSP) isolates were the bacterial strains used in this study. The isolates were cultured on the sheep Blood [local vendor; Ganendra] (8%) Agar Plate (sBAP) followed by incubation at 37˚C with 5% $CO_2$ for 20 hours. The MDR strains used in this study are defined as isolates that are resistant to more than 5 antibiotics.

## Plant extraction

Plant extraction was done by sequential maceration as reported previously [21] with slight modification. Briefly, every 10 g of plant powder was extracted with 100 ml of solvents followed by overnight agitation at 150 rpm. The extraction was done through the following order: n-Hexane extraction, ethyl acetate extraction, ethanol extraction and the final extraction, aqueous extraction. The supernatant obtained from overnight agitation was filtered by Whatman paper. The filtrate was evaporated using a rotary evaporator to obtain solvent-free of plant extracts. The plant powder residue on the Whatman paper was used in the next extraction with the following solvent. Pasta and powder form extracts were harvested from evaporation process.

## Plant extract preparation

Solvent-free plant extracts were prepared at a final concentration of 200mg/ml by dissolving the plant extracts in DMSO. A sterile 6 mm blank disc (Oxoid) was impregnated with 6 mg of plant extract by transferring 30μl of 200mg/ml plant extracts into sterile blank disc [22]. Extracts used for minimum inhibitory concentration (MIC) and minimum bactericidal

concentration (MBC) measurement were prepared in 2-fold dilution starting from 100 mg/ml to 0,4 mg/ml.

## Antibacterial activity determination

**Disc diffusion.** Direct impregnation was used to identify antibacterial activity of extracts using disc diffusion as published previously [22–24] with slight modification in the concentration (6 mg/disc) of extracts used. Briefly, a 0.5 McFarland bacterial suspension of *S. pneumoniae* was prepared in Mueller-Hinton Broth. The suspension was then spread evenly using a sterile cotton swab onto Mueller-Hinton blood (5%) agar. Impregnated disc was then placed on the agar media with control as follows: blank disc impregnated with 100% DMSO and 10% DMSO were used as solvent control while vancomycin 30μg was used as a positive control considering the majority of *S. pneumoniae* are susceptible to vancomycin 30μg with an inhibition zone of $\geq$ 17 mm. The media were then incubated for 20 hours at 37°C with 5% $CO_2$. Inhibition zone diameter observed in the following day was measured to interpret antimicrobial activity. Extracts with inhibition zone > 15 mm were used for MIC and MBC measurements and were tested for antibacterial activity against MDR strain.

**MIC determination.** Microdilution was performed to determine MIC value using a 96-well round bottom plate. The microdilution was performed at volume of 100μl into each well. Concisely, 0.5 McFarland of *S. pneumoniae* suspension was prepared in 5 ml Mueller-Hinton broth. One-hundred microliter of bacterial suspension was then transferred into 11ml of lysed-horse blood [25]. A total 90μl of bacterial suspension in lysed-horse blood was dispensed into each well. From 2-fold diluted extracts tubes, 10μl of extracts was mixed with 90μl of bacterial suspension in each well from column 1–9. Meanwhile, column 10 was used as a positive control by mixing 10μl of vancomycin 4 μg/ml into the 90μl of bacterial suspension. Column 11 was used as a solvent control by mixing 10μl of 10% DMSO or 100% DMSO (ethyl acetate extracts only) into the 90μl of bacterial suspension while column 12 was used as a negative control by addition of 10μl of MHB into the 90μl of *S. pneumoniae* suspension. The vancomycin 4 μg/ml was used as positive control because the majority of *S. pneumoniae* are susceptible to vancomycin 4 μg/ml including the MDR strain. Therefore, the use of vancomycin 4 μg/ml was sufficient to be used as positive control for MIC testing with the ATCC control or the MDR strain used in this study. The plate was then sealed and incubated at 37°C for 20 hours. To produce accuracy in MIC interpretation, a mixture between diluted extracts and lysed-horse blood with the same concentration was used as background control for extract dilution. The MIC value was interpreted as the minimum concentration showing no growth indicated by no turbidity observed in the well.

**MBC determination.** The MBC was determined by inoculating 20μl of mixed suspension in each well of diluted extracts (in the same row) onto a sBAP followed by incubation at 37°C with 5% $CO_2$. The MBC value was defined as the minimum concentration showing no bacterial growth on plate.

**Time-killing measurement.** Time-killing measurement was performed on the ATCC 49619 and MDRSP 2506. It was done by sub-culturing bacterial suspension mixed with plant extract at a certain time of incubation as described previously [26]. The concentrations used for time-killing measurement were 1×MIC, 2×MIC, and 10×MIC. Briefly, 0.5 McFarland of bacterial suspension was prepared in 5 ml MHB. A 900μl of bacterial suspension was then aliquoted into test tubes. A 100μl of extract was added into the bacterial suspension at a final concentration of 1×MIC, 2×MIC, and 10×MIC. Vancomycin of final concentration 4 μg/ml was used as a positive control while bacterial suspension with no addition was used as a negative control. Mixtures were then incubated at 37°C with 5% $CO_2$. Total plate count was done at 0,

1, 3, 6 and 20 hours of incubation. A 10-fold dilution of suspension was done before 100μl inoculation onto sBAP 5%. The CFU/ml was calculated using the following formula

$$\frac{\textbf{Number of colony observed} \times \textbf{dilution}}{\textbf{inoculated volume(ml)}}\,(\textbf{CFU/ml})$$

## Mechanism of action measurements

**Bacterial lysis.** The ATCC 49619 and MDRSP 2506 were used for all mechanism of action testing used in this study. Moreover, *L. inermis* ethyl acetate extract was used for further mechanism of action analysis. Bacterial lysis was determined by measuring absorbance at 595 nm [21] using Varioskan LUX (Thermo). Briefly, a 0.5McFarland of bacterial suspension was prepared in MHB. A volume of 90 μl of this suspension was then transferred into each well of a 96-well flat bottom plate. A 10μl of plant extract was mixed with the 90μl of bacterial suspension at a final concentration of 1×MIC, and 2×MIC. Vancomycin of final concentration 4μg/ml was used as a positive control while bacterial suspension with no addition was used as a negative control. Background control was prepared by mixing 90μl of MHB with 10μl of plant extract at final concentration of 1×MIC, and 2×MIC. All tested groups were performed in triplicates. The plate was then sealed with optical adhesive seal followed by incubation in the Varioskan LUX that is set for kinetic loop with moderate shaki shaking at 60 rpm for every 5 minutes and 10 minutes without shaking. The plate was incubated for 20 hours and absorbance readings at 595 nm was collected every 1 hour.

**Antibiofilm measurements.** Antibiofilm testing was done according to previous publication [27, 28] with slight modification. Briefly, a 50μl of 0.5 McFarland *Streptococcus pneumoniae* was transferred into 5 ml of Brain Heart Infusion supplemented with 1 ml of rabbit serum for enrichment. The enriched suspension was then incubated at 37˚C with 5% $CO_2$ for 5 hours. The bacterial suspension was then adjusted to 0.5McFarland by MHB addition. A 1 ml of the adjusted suspension was transferred into 1.5 ml centrifuge tubes followed by centrifugation at 3000 rpm for 3 minutes. Supernatant was discarded and bacterial cells were washed twice with PBS 1× followed by resuspension with MHB for 1 ml and homogenized. A 5μl of this suspension was transferred into 895μl of lysed-sheep blood. A 100μl of plant extract was mixed with inoculated lysed-sheep blood followed by vortex for homogenization. Final concentrations of plant extract used for antibiofilm measurement were 1×MIC, and 2×MIC of ethyl acetate extract. A 50μl of mixtures was then transferred into 96-well flat bottom plate followed by incubation at 37˚C with 5% $CO_2$ for 20 hours. In the following day, the plate was then washed with PBS 1× for twice followed by tapping on a paper towel and air-dried for 15 minutes. A 50μl of 0.1% crystal violet was transferred into each well followed by incubation for 15 minutes. The plate was washed twice with PBS 1× and was then air-dried for 15 minutes. The crystal violet attached on the bacterial cell wall was then dissolved in 120μl of cold-ethanol absolute and gently mixed. A 100μl of dissolved crystal violet was then transferred into a new 96-well flat bottom plate for absorbance measurement at 594 nm. All the testing were performed in triplicates. Background controls used in this testing consisted of lysed-sheep blood mixed with *L. inermis* extract at a final concentration of 1×MIC, and 2×MIC. Blank control was prepared with lysed-sheep blood and positive control was prepared with mixtures of bacterial suspension in lysed-sheep blood and mouthwash. Negative control was prepared with bacterial suspension in lysed-sheep blood.

## Gene expression changes

**Pre-treatment and RNA extraction.** Before RNA extraction, bacterial cell was treated with plant extract for 1 hour with 2×MIC as the final concentration of the extract in MHB.

Briefly, 1.0 McFarland of bacterial suspension in 5 ml MHB was mixed with *L. inermis* ethyl acetate extract at a final concentration of 2×MIC. The mixture was then incubated for 1 hour. The RNA extraction was done by following the manufacturer's protocol using the Trizol kit (Zymoresearch) with pre-enzymatic lysis buffer. Pre-enzymatic lysis buffer was done following these steps: 1 ml of incubated bacterial suspension was transferred into 1.5 ml micro centrifuge tube followed by centrifugation at 10000 rpm for 2 minutes. Supernatant was discarded and bacterial pellet was resuspended in 180μl pre-enzymatic lysis buffer containing 250U/ml muta-nolysin and lysozyme 20mg/ml followed by incubation at 37˚C for 1 hour [29, 30]. The RNA extraction was continued by following the manufacturer's protocol with DNAse treatment. Concentration of RNA was measured using Qubit HS RNA kit.

**Toxin-antitoxin zeta-epsilon expression (*peZT* and *peZA* genes) and autolysin protein (*lytA*).**   The expression of zeta-epsilon toxin-antitoxin was measured by 2 step qRT-PCR using SYBR Green and following primer pairs: 1050-F (5'-GTC AGA AGT TTA ATG TAT CTT ATG TCG-3'), primer reverse 1050-R (5'-CTC TAA CAT ACG TTC AAT TCC ATC C-3'), primer forward 1051-F (5'-CCA AGA TTA TAC TGA TAG TGA ATT CAA AC-3') primer reverse 1051-R (5'-CTT GCT GCA GTT CTA AAT AGT GTG-3'). Primer pair used for *lytA* detection was *lytA* forward *lytA*-F(5'-ACGCAATCTAGCAGAT-GAAGCA-3') reverse *lytA*-R (5'-TCGTGCGTTTTAATTCCAGCT-3') [31]. The master mix for cDNA synthesis was prepared as the following: nuclease-free water, RT-PCR buffer 1×, reverse transcriptase enzyme and RNA template 1ng/μl. The PCR condition was set up as follows: annealing at 25˚C for 10 minutes, followed by cDNA synthesis at 42˚C for 30 minutes, and inactivation at 85˚C for 5 minutes then stored at 4˚C. The mRNA detection was done by qPCR targeting *peZA*, *peZT*, dan *lytA* gene using master mix as follows: SYBR Green 1×, nuclease-free water, $MgCl_2$ 3mM, primer 0.25μM, and cDNA template 7.5 ng. The PCR was set as follows: pre-denaturation 95˚C 30 seconds, 45 PCR cycles following these conditions: denaturation 95˚C for 2 seconds, annealing 56˚C for 5 seconds and elongation at 72˚C for 13 seconds followed by a dissociation curve. Relative expression was calculated by comparing the Ct value with a calibrator.

**Ultrastructure changes.**   Ultrastructure changes was observed under a transmission electron microscope. Before the bacterial cells were processed for blocking, pre-treatment of bacterial cells with plant extract was done at a final concentration of 2×MIC of *L. inermis* ethyl acetate extract as described previously [21] with slight modification. The MDRSP 2506 and ATCC 49619 were the strains used for the ultrastructure changes observation. Negative control used was a bacterial suspension without any additional treatment. Concisely, a 0.5 McFarland bacterial suspension was prepared in BHI broth supplemented with rabbit serum. The suspension was then incubated for 6 hours at 37˚C with 5% $CO_2$. *Lawsonia inermis* ethyl acetate extract was added into the bacterial suspension at a final concentration of 2×MIC. Mixtures were re-incubated for 2 hours at 37˚C with 5% $CO_2$. Then, 1 ml of treated bacterial suspension was transferred into 1.5 ml microtubes followed by centrifugation at 3000 rpm for 3 minutes. The supernatant was then discarded. This procedure was repeated for 5 times following the total volume of bacterial suspension. Bacterial suspension was then washed with PBS 1× for 3 times. Fixation was performed with 1 ml of 2.5% glutaraldehyde in 0.1M cacodylate and 3% sucrose followed by gentle homogenization for 15 minutes then stored at 4˚C for 48 hours.

Block preparation was prepared following previous publication [32] using glutaraldehyde fixative solution. The procedure starts with washing the bacterial cells with 1 ml of 0.1M Cacodylate buffer, then vortex with low speed for 15 minutes in 4˚C followed by centrifugation at 3000 rpm for 3 minutes. This washing process was repeated for two times. After washing the bacterial cells, the cells were resuspended in fixation buffer consisting of 2% osmium tetroxide and 2.5% $K_3Fe (CN)_6$ for 1 hour at 4˚C with low speed of vortex. Bacterial cells were then

washed with 1 ml of 0.1M Cacodylate buffer three times. After fixation procedures, the bacterial cells were processed for dehydration by serial procedures starting from 50% ethanol for 15 minutes, 70% ethanol for 18 hours, 80% ethanol for 15 minutes, 96% ethanol for 15 minutes and final dehydration was done using ethanol absolute for 15 minutes two times. At the end of each dehydration step, the bacterial cell was processed with low vortex in 4˚C followed by centrifugation at 3000 rpm for 3 minutes. Infiltration with propylene oxide was then done two times and incubated in 4˚C for 15 minutes and centrifugation at 3000 rpm for 3 minutes after serial dehydration. Embedding process was done with spurr's resin added gradually in ratio with propylene oxide as follows spurr's resin: propylene oxide (1:2), spurr's resin: propylene oxide (1:1), spurr's resin: propylene oxide (2:1), and pure spurr's resin. Each process was done for 15 minutes at room temperature while pure spurr's resin was done for 2 hours in 500μl tubes followed by incubation in vacuum for 20 hours. Spurr's resin was harden in oven at 56˚C for 48 hours. A harden spurr's resin could be used for trimming and processing. Triple lead citrate was used as dye prior to observation under TEM. The samples were observed with 8000–12000 magnifications.

**Phytochemical compounds detection.** The phytochemical compounds were detected using *Liquid Chromatography–Mass Spectrophotometry* (LC-MS/MS). A 1 mg/ml of *L. inermis* extract was prepared in sterile water prior to detection. Detection was done at Pusat Laboratorium Forensik, Sentul, Bogor. Spectra obtained by LC-MS/MS machine were then analyzed by using MassLynx v4.1 software to define what molecules detected at certain m/z ratio. The molecular formula was then submitted to http://www.chemspider.com for further analysis.

**Data analysis.** All data obtained in this study were numeric data that were analyzed for their normality and homogeneity. All data will be analyzed with parametric statistical test if the data were distributed normally and are homogenous. ANNOVA one-way test will be used for parametric analysis followed by LSD post hoc test. Meanwhile, non-parametric statistical analysis was used if the data were not homogenous and not normally distributed. Kruskall-Wallis and Mann-Whitney U test was done for analyzing non-parametric data.

## Results

### Plant extracts

Among the 13 medicinal plants used in this study, there were 3 extracts not processed for antimicrobial activity testing; *H. sabdarifa* n-hexane extract, *A. paniculata* aqueous extracts and *P. futida* ethanol extracts due to the limited amount extracts obtained after maceration. Pasta form extracts were obtained from n-hexane and ethyl acetate extracts while ethanol and aqueous solvent resulted in powdery extract. *Corypha utan*, Sinsus, and *Vernonia amygdalina* were only extracted with ethyl acetate and ethanol.

### Antibacterial activity

**Disc diffusion.** Disc diffusion was performed as an initial antibacterial activity screening of the extracts, in which, the bacterial activity is measured according to the inhibition zone resulted by the diffusion of compounds into the agar media. Extracts with inhibition zone > 11 mm against *S. pneumoniae* ATCC 49619 were continued for antibacterial testing against MDR strain of *S. pneumoniae*.

The results of the inhibition zone showed that extracts obtained by ethyl acetate and ethanol solvent had stronger antibacterial activity than extracts obtained by n-hexane and aqueous solvent (Fig 1). There was also a significant difference in antibacterial activity that is among solvents according to their inhibition zone (*Kruskal-wallis*; P<0.01). Furthermore, we observed that ethyl acetate and ethanol extracts showed higher antibacterial activity significantly

a.

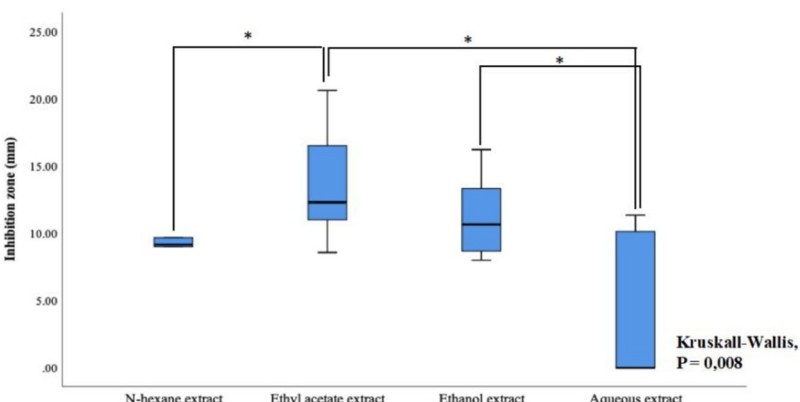

* Mann-Whitney, P<0,05

b.

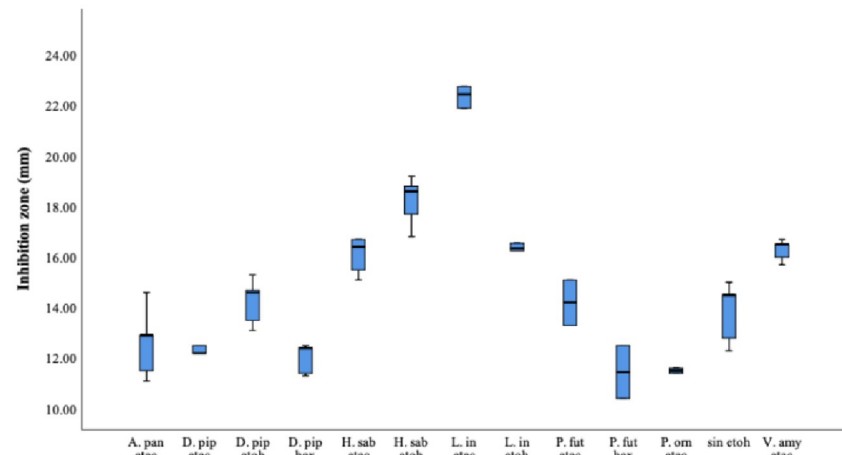

**Fig 1. Comparison of antibacterial activity of plant extracts by disc diffusion.** a) Inhibition zone of extracts on ATCC 49619 in various solvent. b) Inhibition zones of plant extracts on multi-drug resistant strain *S. pneumoniae*. Extract-impregnated disc (6mg/disc) was used to obtain inhibition zone performed on sheep blood (5%) Mueller-Hinton agar. The incubation was done at 37˚C with 5% $CO_2$ for 20 hours. A. pan etac; *Andrographis paniculate* ethyl acetate extract, D. pip etac; *Drymis piperita* ethyl acetate extract, D. pip etoh; *Drymis piperita* ethanol extract, D. pip hex; *Drymis piperita* n-hexane extract H. sab etac; *Hibiscus sabdariffa* ethyl acetate extract, H. sab etoh; *Hibiscus sabdariffa* ethanol extract, L. in etac; *Lawsonia inermis* ethyl acetate extract, L. in etoh; *Lawsonia inermis* ethanol extract P. fut etac; *Passiflora foetida* ethyl acetate extract, P. fut hex; *Passiflora foetida* n-hexane extract, P. orn etac; *Piper ornatum* ethyl acetate extract, Sin etoh; Sinsus ethanol extract, V. amy etac; *Vernonia amygdalina* ethyl acetate extract. *Mann-Whitney U test (P<0,05).

different than n-hexane extracts and aqueous extracts (Mann-Whitney; P<0.01) against ATCC 49619. Meanwhile, there was no significant difference between ethanol and ethyl acetate extracts (Fig 1). Inhibition zones resulted from plant extracts diffusion were observed vary against tested bacteria. The minimum inhibition zone was no inhibition (defined as 0) while maximum inhibition zones were defined as zones measuring at 21 mm. Ethyl acetate and ethanol plant extracts were observed to have had more frequent inhibition zone compared to n-hexane extracts and aqueous extracts (S1 Fig).

According to the disc diffusion screening on ATCC 49619, we decided to include extracts with inhibition zones > 11 mm to be processed against MDR strain of *S. pneumonae*

(MDRSP). The extracts processed for antibacterial screening on MDRSP were *A paniculata* ethyl acetate extract, *D. piperita* ethyl acetate extract, *D. piperita* ethanol extract, *D. piperita* n-hexane extract, *H. sabdarifa* ethyl acetate extract, *H. sabdarifa* ethanol extract, *L. inermis* ethyl acetate extract, *L. inermis* ethanol extract, *P. futida* ethyl acetate extract, *P. futida* n-hexane extract, *P. ornatum* ethyl acetate extract, sinsus ethanol extract and *V. amygdalina* ethyl acetate extract. Disc diffusion showed various inhibition zone ranging from 11 mm to 22 mm (Fig 1). According to the results, *H. sabdarifa* ethyl acetate and ethanol extracts, *V. amygdalina* ethyl acetate extract, as well as *L. inermis* ethyl acetate and ethanol extracts showed inhibition zones >15 mm. These extracts were selected for MIC and MBC determination on ATCC 49619 and MDRSP with microdilution.

**MIC and MBC measurements.** The microdilution showed that the *L. inermis* ethyl acetate had the lowest concentration of MIC and MBC (0.16 mg/ml) followed by *V. amygdalina* (0.63 mg/ml). Other extracts showed MIC and MBC value of more than 1 mg/ml (MIC range 1.25–2.5 mg/ml and 2.5 mg/ml for MBC) against ATCC 49619 (Fig 2). Moreover, we also tested some extracts previously showing inhibition zones <15 mm on MDRSP for confirmation such as *D. piperita* ethanol extract (Ø = 14 mm), *H. sabdarifa* aqueous extract (Ø = 11 mm), *A. paniculata* ethyl acetate extract (Ø = 12 mm), *P. ornatum* ethyl acetate extract (Ø = 12 mm), and *P. futida* ethyl acetate extract (Ø = 14 mm). These extracts showed MIC value of more than 1 mg/ml (range 2.5–10 mg/ml) and MBC value range 2.5 –>10 mg/ml (S1 Table). Plant extracts with MIC value <1 mg/ml were selected for testing on MDRSP. Therefore, the *L. inermis* and *V. amygdalina* ethyl acetate extracts were chosen for testing on MDRSP with *L. inermis* ethanol extract as a comparison for representative extracts with MIC value above 1 mg/ml. Among these extracts, the *L. inermis* ethyl acetate extract showed the lowest MIC value against MDRSP; ($\bar{x}$ = 0.22 mg/ml; range 0.16–0.31 mg/ml) followed by *V. amygdalina* ethyl acetate extract; ($\bar{x}$ = 0.57 mg/ml; range 0.31–0.63 mg/ml). Ethanol extract of *L. inermis* showed an MIC value ($\bar{x}$) = 0.878 mg/ml ranging from 0.63 to 1.25 mg/ml. *Lawsonia inermis* ethyl acetate extract also had the lowest MBC value; ($\bar{x}$ = 0.25 mg/ml; range 0.16–0.31 mg/

| No. | Extract | Solvent | Concentration of Extracts (mg/ml) | |
|---|---|---|---|---|
| | | | MIC | MBC |
| 1 | *H. sabdariffa* | Ethyl acetate | 1,25 | 2,5 |
| | | Ethanol | 2,5 | 2,5 |
| 2 | *L. inermis* | Ethyl acetate | 0,16 | 0,16 |
| | | Ethanol | 1,25 | 2,5 |
| 3 | *V. amygdalina* | Ethyl acetate | 0,63 | 0,63 |

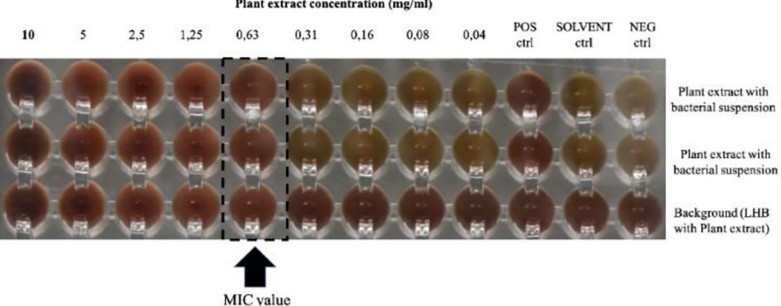

**Fig 2. Minimum inhibitory concentratio of plant extracts on ATCC 49619 and Multi-drug resistant strain *S. pneumoniae*.** Extract-impregnated disc (6mg/disc) was used to obtain inhibition zone performed on sheep blood (5%) Mueller-Hinton agar. The incubation was done at 37˚C with 5% $CO_2$ for 20 hours. The MIC values were obtained by microdilution using plant extract concentration ranged from 10–0,04 mg/ml with vancomysin 4µg/ml used as positive control.

ml) while *V. amygdalina* showed an MBC value (x̄) = 0.57; range 0.31–0.63 mg/ml. *Lawsonia inermis* ethanol extract had the highest MBC value compared to other extracts; (x̄) = 3 mg/ml ranging from 2.5 to 5 mg/ml. According to the MIC results, we decided to use *Lawsonia inermis* ethyl acetate extract for determining the mechanism of action of *L. inermis* extracts against *Streptococcus pneumoniae* ATCC 49619 and MDRSP 2506. The MDRSP 2506 was selected because this isolate was confirmed as a multi-drug resistant, resistant to 9 antibiotics including Penicillin, Meropenem, Azithromycin, Tetracycline, Erythromycin, Cefuroxime, Amoxicilin: Clavulanic Acid (2:1), Trimetophrim/Sulfamethoxazole, and Clindamycin.

**Time-killing of *Lawsonia inermis* extract.** *Lawsonia inermis* ethyl acetate extract at a final concentration of 1×MIC, 2×MIC and 10×MIC showed no bacterial growth of ATCC 49619 and MDRSP 2506 in the early hour of incubation with the extracts (Fig 3a and 3b respectively). In comparison with vancomycin 4 µg/ml, this antibiotic showed a decline of viable bacteria overtime with total killing was observed at 20 hours of incubation. Meanwhile, the negative control showed bacterial growth until 20 hours of incubation. Bactericidal activity was observed at the first hour of incubation of *Lawsonia inermis* ethyl acetate extract, indicated by total killing of bacterial cells over time during incubation.

## Mechanism of actions

**Bacterial lysis.** *Lawsonia inermis* ethyl acetate extract was observed to cause bacterial lysis on *Streptococcus pneumoniae* ATCC 49619 and MDRSP 2506 as shown by the decrease in absorbance at 595 nm over time. Bacterial lysis activity was observed to have started early during the first hour of incubation with *L. inermis* ethyl acetate extract. *Streptococcus pneumoniae* treated with vancomycin 4 µg/ml also showed bacterial lysis activity during the early hour of incubation. Meanwhile, the negative control showed a logarithmic curve followed by a plateau and an eventual decline, depicting a normal bacterial growth curve (Fig 4a and 4b).

**Antibiofilm formation activity.** *Lawsonia inermis* ethyl acetate extract showed antibiofilm activity against ATCC 49619 and MDRSP 2506 that is statistically significant compared to the negative control (Fig 4c). The extract showed strong inhibition against ATCC 49619 and MDRSP 2506. The biofilm inhibition was shown by a lower absorbance at 594 nm, meaning that less crystal violet is binding on the bacterial cell wall that formed biofilm at bottom of the well. Percent inhibitions of *L. inermis* ethyl acetate extract against ATCC 49619 were 90% for 1×MIC and 96% for 2×MIC showing that increase in plant extract concentration was followed

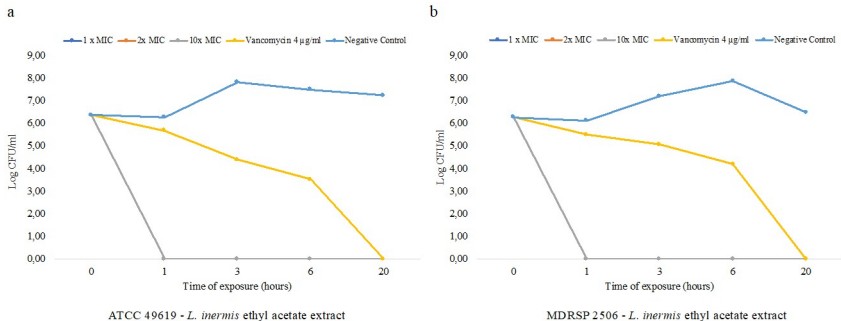

**Fig 3. Time-killing curves over time of incubations of *Lawsonia inermis* ethyl acetate extract.** a) *S. pneumoniae* ATCC 49619, b) MDRSP 2506. Time-killing assay was performed by direct platting with total plate count to measure viable cell during plant extract exposure at 0, 1 3, 6 and 20 hours of exposures. *Lawsonia inermis* final concentrations at 1×MIC, 2×MIC and 10×MIC were used for incubation, vancomycin at final concentration 4 µg/ml was used as growth control under antibiotic exposure while untreated cells were used as negative control. Curve 2×MIC is overlapped with curve 10×MIC showing similar effect of killing on *S. pneumoniae*.

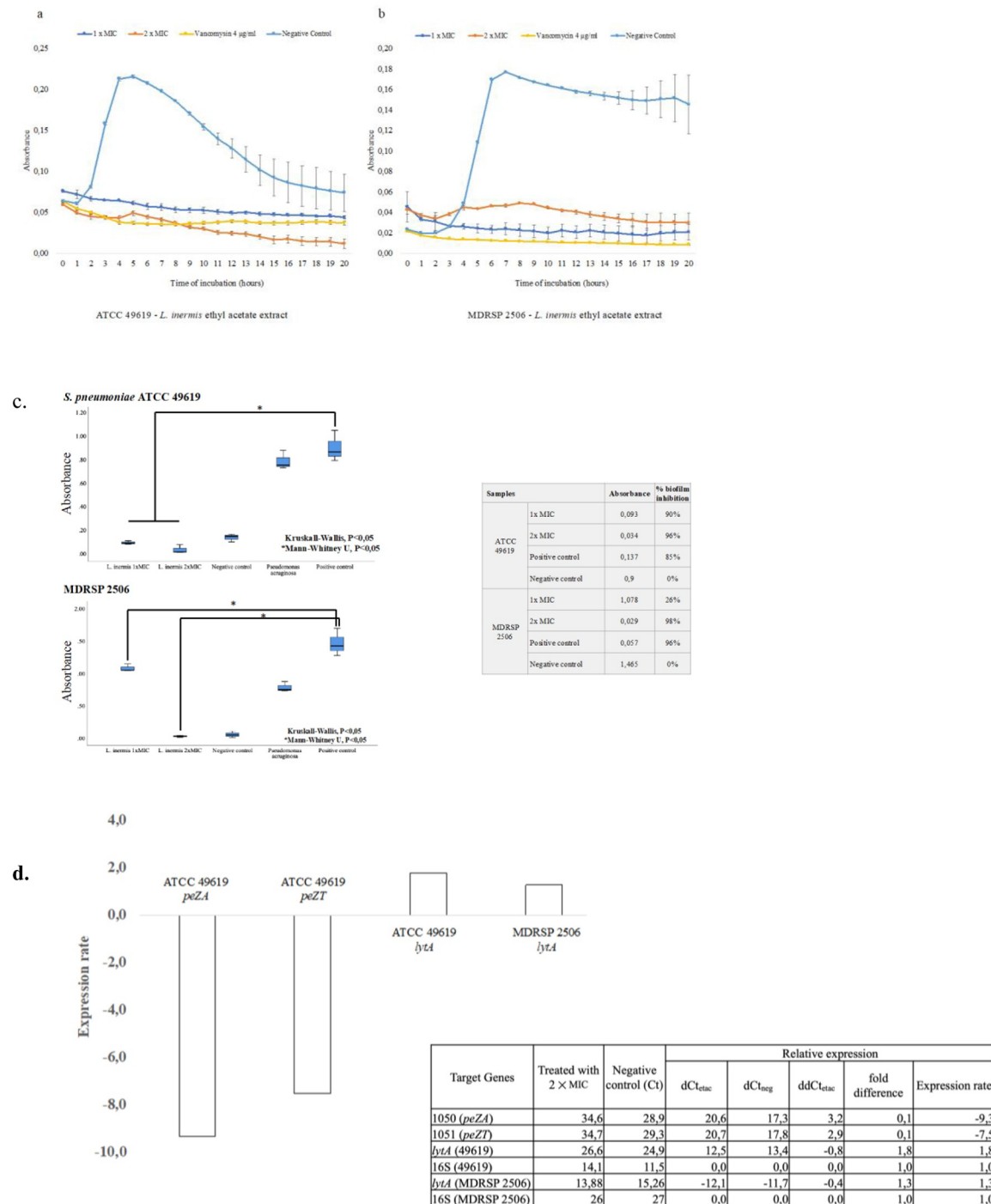

**Fig 4. Mechanism of action of *Lawsonia inermis* ethyl acetate extract on *Streptococcus pneumoniae*.** a) Bacterial lysis curves of *S. pneumoniae* ATCC 49619, b) Bacterial lysis curves of MDRSP 2506, c) antibiofilm activity, d) Expression rate of *peZA*, *peZT*, and *lytA*. Bacterial lysis measurement was performed by measuring absorbance of 595 nm each 1 hour of 20 hours of incubation. *Lawsonia inermis* final concentrations at 1×MIC, 2×MIC and 10×MIC were used for incubation, vancomycin at final concentration 4 µg/ml was used as growth control under antibiotic exposure while untreated cells were used as negative control. Antibiofilm activity was measured by using 1% crystal violet for detection. Initial enrichment was done in BHI supplemented with rabbit serum incubated for 5 hours. Lysed-sheep blood was used for inducing biofilm formation performed in 96-well plate flat bottom. PBS 1× was used for plate washing and 0,5% crystal violet was used to detect biofilm formed in each well. *Lawsonia inermis* final concentrations at 1×MIC, and 2×MIC were used for inhibition testing, mouthwash at final concentration 1:10 was used as inhibition control while untreated cells were used as negative control. Prior to RNA extraction, the pathogen (1 McFarland) was incubated with 2×MIC of *L. inermis* ethyl acetate extract for 1 hours. RNA concentration was measured by RNA Qubit HS kit. Expression rate was measured by two-steps qRT-PCR.

by higher antibiofilm activity. The lowest inhibition activity was shown by 1×MIC against MDRSP 2506 that resulted in 26% biofilm formation inhibition while the 2×MIC concentration showed stronger inhibition (98%). In comparison, mouthwash was used as a control and showed 85% and 96% inhibition against ATCC 49619 and MDRSP 2506 respectively.

**Epsilon-zeta and autolysin expression.** The RNA concentration obtained by the Trizol kit RNA extraction showed varying concentration of RNA between untreated cells and treated cells with *L. inermis* extract. Treated cells with *L. inermis* ethyl acetate extract showed less RNA (ATCC 49619, $\bar{x} = 1.005$ ng/$\mu$l; MDRSP 2506, $\bar{x} = 1.83$ ng/$\mu$l) relative to untreated cells (ATCC 49619, $\bar{x} = 4.88$ ng/$\mu$l; MDRSP 2506, $\bar{x} = 3.805$ ng/$\mu$l).

The qRT-PCR showed that ATCC 49619 and MDRSP 2506 expressed autolysin protein encoded by *lytA* gene. However, only ATCC 49619 expressed zeta-epsilon toxin-antitoxin system. Therefore, analysis on gene expression change of zeta-epsilon toxin-antitoxin system was done only on ATCC 49619. According to the Livak method, relative expression of antitoxin protein, epsilon; encoded by *peZA*, and toxin protein zeta; encoded by *peZT*, of ATCC 49619 was decreasing 9.3-fold for the epsilon protein (*peZA*) and 7.5-fold for the zeta toxin protein (*peZT*) normalized with a calibrator. Meanwhile, we found that the expression of *lytA* increased 1.8-fold for ATCC 49619 and 1.3-fold for MDRSP 2506. The 16SrRNA was used as a reference gene (Fig 4d).

**Ultrastructure changes.** Under a Transmission Electron Microscope, we discovered some ultrastructural changes occurring on the bacterial cell of *S. pneumoniae* after treatment with *L. inermis* ethyl acetate extract. The normal morphology of a *S. pneumoniae* cell was observed as diplococcus commonly in lancet form morphology. Other morphological characteristics of a normal *S. pneumoniae* (untreated cells) also include the existence of a pilus, a pneumococcal capsule, and a clear cell wall structure (Fig 5a). The *L. inermis* ethyl acetate extract have resulted in some ultrastructural changes in bacterial morphology and cell appearance of MDRSP 2506 (Fig 5b) and ATCC 49619 (Fig 5c–5e) with ATCC 49619 showing more changes than MDRSP 2506. The MDRSP 2506 treated with the *L. inermis* extract showed long-chain formation (Streptococcus) as the commonest morphology instead of a typical

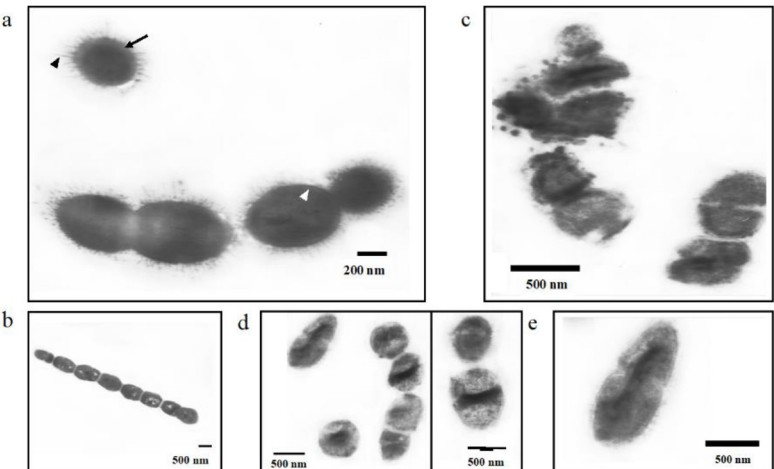

**Fig 5. Ultrastructure changes of *S. pneumoniae* affected by *L. inermis* extract.** a) untreated *S. pneumoniae* showing normal ultrastructure of *S. pneumoniae*. b-e) treated *S. pneumoniae* with *L. inermis* extract showing various ultrastructure disorder on *S. pneumoniae*. Prior to sample block preparation for TEM, the pathogen (1 McFarland) was incubated with 2×MIC of *L. inermis* ethyl acetate extract for 2 hours. The cells were then fixed with 2,5% glutaraldehyde. Cells were blocked in Spurr's resin. Ultrastructure changes were observed under 8000–12000 magnification. Black triangle: pilus structure, black arrow head: capsule, white arrow: cell wall.

diplococcus as normal *S. pneumoniae*. In addition, cell wall disorder was also observed indicated by invagination of the cell wall in some bacterial cells with no more lancet form observed. The appearance of nucleoid condensation and empty area inside the bacterial cell were also observed on the MDRSP 2506 treated with *L. inermis* ethyl acetate extract. The MDRSP 2506 treated with the extract also did not express a pilus compared to the non-treated cells (S2b Fig). All of ultrastructure disorders were not observed on the non-treated bacterial cells (S2a Fig).

The ATCC 49619 treated with the *L. inermis* ethyl acetate extract showed more varying changes compared to the MDRSP 2506 (S5 Fig). The addition of *L. inermis* ethyl acetate extract caused bacterial lysis indicated by cellular destruction and presence of extracellular vesicle. Moreover, cell wall destruction or incomplete cell wall structure was clearly observed on the treated cells compared to non-treated bacterial cell (S3a, S3b and S4b Figs). Appearance of protoplast; bacterial structure without cell wall, was very common on the ATCC 49619 with extract compared to the normal bacteria, non-treated bacteria. Furthermore, a hole on the cell wall was observed in the ATCC 49619 treated with extract, showing that cell wall integrity was disturbed resulting (S3b Fig). Another ultrastructural change observed on the treated ATCC 49619 was cell membrane folding, causing a space between cell wall and cell membrane showing that cell membrane might lose its integrity (S3b and S4b Figs). Appearance of the treated ATCC 49619 cytoplasm was also observed to have faded in color compared to the normal, non-treated bacterial cell that is showing a darker cytoplasm. Cytoplasm condensation was also observed on the treated ATCC 49619 indicated by a darker region in the middle of the cell which was not observed on the normal, non-treated cell (S4 Fig).

**Phytochemical compounds of *L. inermis*.**   Spectra obtained from LC-MS/MS showed various peaks observed during 22 minutes of reading. Peaks were observed mostly during the early 5 minutes and between the last 7 minutes of time of flight (Fig 6). Among the spectra analyzed with the MassLynx v.4.1, we found that phenolic compounds were the most prevalent compound observed in ethyl acetate extracts of *Lawsonia inermis* leaves extract. Besides phenol benzene derivate compounds, there were also some quinone and chromone derivate compounds observed in this extract. There were some undetermined compounds observed during analysis with the MassLynx v.4.1 and http://www.chemspider.com. However, we provided the formula of some undetermined molecules observed in this study (S2–S4 Tables).

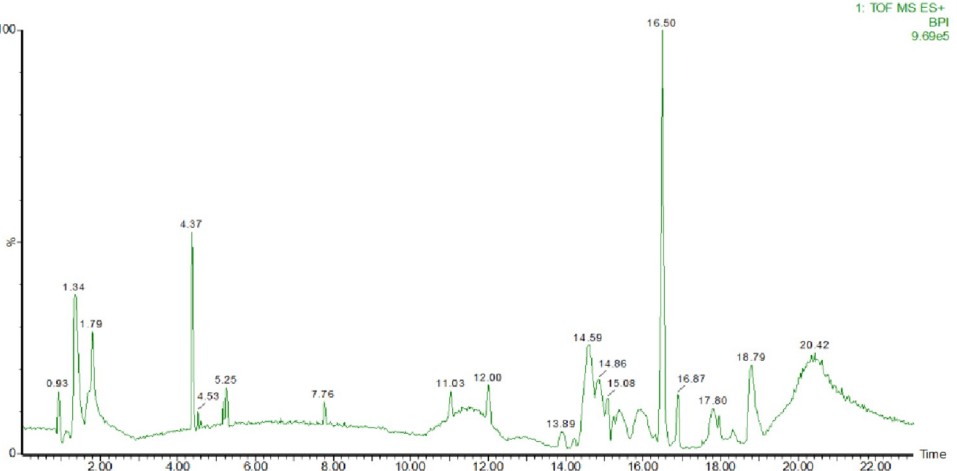

**Fig 6. Chromatogram of *L. inermis* ethyl acetate extract obtained with LC MS/MS.** A 1 mg/ml plant extract diluted in sterile ddH$_2$O was used for phytochemical compound detection with LC MS/MS.

## Discussion

This study discovered that ethyl acetate, and ethanol solvent showed higher antibacterial activity compared to n-hexane and aqueous. There were many previous studies reporting that the use of ethyl acetate and ethanol resulted in higher antibacterial activity against pathogenic bacteria compared to hexane, chloroform [33–35], aqueous [34, 36], and other solvents; methanol [33, 35, 36], and butanol [35]. when comparing the phytochemical compounds isolated from each solvent, ethyl acetate and ethanol solvent could extract more phytochemical compounds rather than n-hexane, chloroform [33–35]. Other than ethyl acetate and ethanol solvent, aqueous extract was also reported to have higher amount of phenolic, flavonoids, and tannins compound in *Punica granatum* compared to chloroform and hexane extract [34]. A study of *Commelina nudiflora* reported that phenolic and flavonoid compounds were highly extracted in aqueous and ethanol solvents compared to hexane, chloroform and ethyl acetate [37]. These reports have shown that solvents isolating more phytochemical compounds may produce higher antibacterial activity.

First, antibacterial screening of medicinal plant extracts was done with disc diffusion against *S. pneumoniae* ATCC 49619 followed by screening on MDRSP strain. The categorization used in determining antibacterial activity is based on the inhibitory zone diameter [38–40]: inhibition zone diameter >20 mm was categorized as very strong antibacterial activity, 15–20 mm as strong, 10–14 mm as moderate and 9–7 mm as weak, while <7 mm as no antibacterial activity. According to this category, n-hexane plant extracts showed no antibacterial activity to strong against ATCC 49619 with *D. piperita* showed strong antibacterial activity (Ø = 21 mm) while weak activity was a dominant category in n-hexane plant extract. Meanwhile, ethyl acetate plant extract also showed no activity to very strong antibacterial activity against *S. pneumoniae* ATCC 49619 with *V. amygdalina* extract showing very strong antibacterial activity. Moderate and strong antibacterial activities categories were the commonest category in ethyl acetate extract. Similar with n-hexane extract, ethanol plant extract also showed antibacterial activity ranging from no activity to strong antibacterial activity with moderate activity as the most common category. *Lawsonia inermis* and sinsus showed strong antibacterial activity against *S. pneumoniae* ATCC 49619. In addition, aqueous extract showed no antibacterial activity to moderate with no activity as the most prevalent category.

In addition, antibacterial activity against MDRSP strain was also reported to vary, ranging from moderate to very strong antibacterial activity with *Lawsonia inermis* ethyl acetate extract showing very strong antibacterial activity (Ø = 22 mm). *Hibiscus sabdarifa* ethyl acetate and ethanol extracts showed strong antibacterial activity against MDRSP strain with an inhibitory zone diameter of 17 and 18 mm respectively. The *L. inermis* ethanol extract and *V. emygdalina* ethyl acetate extract also showed strong antibacterial activity against MDRSP strain with inhibitory zone diameter of 16 mm for each.

Furthermore, *H. sabdarifa* ethyl acetate and ethanol extracts, *L. inermis* ethyl acetate and ethanol extracts and *V. amygdalina* ethyl acetate extracts were tested for MIC and MBC determination through with microdilution followed by agar plate inoculation. Interpretation of the MIC value obtained by microdilution was done according to a previously established range category as follows: highly active antibacterial was defined as MICs value <0.1 mg/ml; MICs value ranging between 0.1mg/ml and 0.5 mg/ml were defined as active; MICs value ranging in between 0.5 mg/ml and 1mg/ml were defined as moderate; MICs ranging from 1 mg/ml to 2 mg/ml were considered as low activity while > 2 mg/ml was considered as no antibacterial activity [41]. According to these categories, we defined *H. sabdarifa* ethyl acetate extract as low antibacterial activity (MIC value = 1.25 mg/ml) while the ethanol extract showed no antibacterial activity (MIC value = 2.5 mg/ml) against *S. pneumoniae* ATCC 49619. Ethyl acetate extract

of *V. amygdalina* showed moderate activity (MIC value = 0.63 mg/ml) against *S. pneumoniae* ATCC 49619. Although the ethanol extract of *L. inermis* only showed low antibacterial activity (MIC value = 1.25 mg/ml), the ethyl acetate *L. inermis* extract was defined however as active (MIC value = 0.16 mg/ml) in antibacterial activity against *S. pneumoniae* ATCC 49619.

In addition, among all tested plant extracts, bactericidal activity was observed in 2-fold concentration of MIC value on *H. sabdarifa* ethyl acetate extract and *L. inermis* ethanol extract (MBC value = 2.5 mg/ml for each). Meanwhile, *H. sabdarifa* ethanol extract, *V. amygdalina* ethyl acetate and *L. inermis* ethyl acetate extract showed an MBC value that is similar with the MIC value; 2.5 mg/ml, 0.63 mg/ml and 0.16 mg/ml respectively. These MBC concentrations are needed to kill 99.9% of the bacteria after 24 hours of incubation [42]. The MBC value is of considerable importance in antibacterial agent use on patients with vital organ infection such as endocarditis and meningitis infection. By considering the pharmacodynamic, pharmacokinetic and the MBC value of the antibacterial, dosage of the antibacterial agent with bactericidal activity can be formulated for patients with vital organ infection patients. This bactericidal concentration is necessary to avoid relapse infection in vital organ [43–45].

In comparison to previous study, *H. sabdariffa* was reported showing weak to moderate activity against oral pathogenic bacteria such as *Streptococcus mutans*, *Streptococcus sanguinis*, *Capnocytophaga gingivalis*, and *Staphylococcus aureus* based on their inhibitory zone diameter. However, the MIC and MBC value showed 5 mg/ml and 20 mg/ml as the lowest concentration, respectively, against these pathogenic bacteria [46]. Moreover, 100 mg/ml of *V. amygdalina* was also reported having strong antibacterial activity against *Vobrio* sp. The same concentration also resulted in moderate antibacterial activity against *Serratia* sp., *Micrococcus* sp., and *Enterococcus* sp based on their inhibitory zone diameter [47]. The 200 mg/ml ethanolic extract of *V. amygdalina* was also reported to have strong antibacterial activity against *Eschericia coli* and *P. aeruginosa* based on their inhibition zone diameter. Moderate activity was also reported using the same concentration against *Klebsiella pneumoniae*, and *Bacillus subtilis* while weak activity was observed against *S. aureus* [48]. The *Lawsonia inermis* ethanol extract was reported having very strong antibacterial activity against *P. aeruginosa* (inhibition zone diameter >20 mm). Moreover, *Lawsonia inermis* also showed antibacterial activity against some pathogenic bacteria such as *Staphylococcus epidermidis*, *S. aureus*, *E. coli*, *K. pneumoniae*, *Salmonella* sp., *Shigella* sp., *Vibrio cholerae*, *Neisseria meningitidis*, *Haemophilus influenzae*, *Streptococcus pyogenes* and *S. pneumoniae*. The antibacterial activity was reported to vary over the region of *L. inermis* was collected. A previous study reported antibacterial activity of *L. inermis* against *S. pneumoniae* ranging from moderate to very strong activity [18]. Another study also reported that *Lawsonia inermis* aqueous and ethanol extract showed active antibacterial activity (MIC value = 0.125 mg/ml) against *S. aureus*, *P. aeruginosa*, *Candida albicans* [49] and *S. epidermidis* [19].

In this study, we also measured the time-killing of *Lawsonia inermis* ethyl acetate extract against *S. pneumoniae* ATCC 49619 and MDRSP 2506. This study found that *L. inermis* ethyl acetate extract showed bactericidal activity at 1 hour incubation resulting in the total killing of viable bacteria. Meanwhile, vancomysin 4 μg/ml showed bactericidal activity after 6 hours of incubation. The bactericidal activity was determined by calculating reduction of viable bacteria overtime. Bactericidal activity was recorded when reduction of viable cell was greater than >$3\log_{10}$ (CFU/ml) where the $3\log_{10}$-fold reduction is equal to 99.9% reduction. Time-killing measurement is used to determine bactericidal activity over time and interaction of antimicrobial agents. The synergistic combination of antimicrobial agent can be determined by the reduction in viable cells of more than $2\log_{10}$ (CFU/ml)-fold and antagonistic combination is determined by the increase in viable cell of more than $2\log_{10}$ (CFU/ml)-fold compared to the most active single agent [26, 50]. A previous study also reported that *L. inermis* showed

bactericidal activity against *S. aureus*, *B. subtilis*, *E. coli*, *Salmonella typhi*, *Klebsiella spp*. *S. epidermidis*, dan *Shigella sonnei* after 12 hours of incubation [51]. Another study also reported bactericidal activity of *L. inermis* on *E. coli* observed after 2 hours of incubation [52].

*Lawsonia inermis* ethyl acetate extract was also found to induce bacterial lysis of *S. pneumoniae* indicated by a decrease in absorbance at 595 nm. The Autolysin protein of *S. pneumoniae*, encoded by *lytA* gene, is involved in bacterial lysis. Activation or overexpression of autolysin is correlated with cell lysis of *S. pneumoniae* and release of DNA into environment in competent cells [53–57]. Overexpression of autolysin due to exposure of *L. inermis* extract might explain one of the causes inducing bacterial lysis of *S. pneumoniae*. In addition, this study found that bacterial lysis was also in concordance with the increase of *lytA* expression after 2 hours of incubation with 2×MIC of *L. inermis* ethyl acetate extract. Therefore, *L. inermis* ethyl acetate extract might induce bacterial lysis on *S. pneumoniae* through autolysin activation pathway. However, details on how the *L. inermis* ethyl acetate also induces competent on *S. pneumoniae* still remains undetermined in this study. A study also reported that plant extract induces various effects on bacteria including bacterial lysis, cell wall disorder, swelling, and cell division anomaly which were reported on *Bacillus cereus* treated with extract of *Buxus macowanii* [58]. Moreover, detection of DNA or RNA released by cells into media was not able to determine bacterial leakage caused by plant extract since we found inconsistent data obtained overtime due to nucleotide degradation (DOI: 10.6084/m9.figshare.20364249).

The ability of *L. inermis* ethyl acetate extract as an antibiofilm formation was also discovered in this study. High antibiofilm activity was observed against *S. pneumoniae* ATCC 49619 ($\geq$ 90% inhibition) and MDRSP 2506 (98% inhibition at 2×MIC concentration). These antibiofilm activity was higher than in previous studies reporting antibiofilm activity of *L. inermis* on *P. aeruginosa* (69.2% inhibition) [59, 60], and MRSA (84.7% inhibition) [61]. Biofilm formation can reduce the efficiency of the antimicrobial against bacterial infection due to the extracellular matrix of the biofilm that limits the drug in reaching the bacterial cells [62, 63]. There are some possible strategies that might explain how an antibiofilm agent can reduce biofilm formation. Inhibition of quorum sensing AHL (N-acyl homoserine lactone) pathway, lipopolysaccharide disaggregation, and inhibition of protein production used for adhesion on surface of object or cells, are some strategies used to inhibit biofilm formation [64]. However, the strategy used by *L. inermis* extract to inhibit biofilm formation on *S. pneumoniae* remains unclear.

In this study, further analysis on toxin-antitoxin system; zeta-epsilon, on *Streptococcus pneumoniae* was done to describe the disorder and changes of bacterial cell ultrastructure. However, MDRSP 2506 did not express the epsilon-zeta protein showing that this isolate was not as virulence as ATCC 49619 that produced epsilon-zeta protein [65]. A previous report also described that *S. pneumoniae* without zeta-epsilon toxin-antitoxin system showed less invasiveness and is rarely reported in causing IPD since *S. pneumoniae* with this toxin-antitoxin system will dominate the competition [66]. Therefore, the analysis for epsilon-zeta was done only for ATCC 49619.

This study discovered that exposure of *L. inermis* ethyl acetate extract at 2×MIC concentration resulted in various changes and disorder such as long-chain morphology, cell membrane folding and cell wall disorder. Epsilon-zeta expression rate analysis showed that there was a decreasing in the expression rate of epsilon; antitoxin protein, and zeta; toxin protein compared to untreated bacterial cells. Epsilon protein showed higher reduction (9.3-fold reduction) in expression compared to the zeta toxin (7.5-fold reduction) resulting in the distribution of a toxic zeta protein within the cell. In addition, epsilon antitoxin protein is an antitoxin type II which is more labile on environmental stress. This protein will degrade (proteolysis) during environment environmental stress such as antibiotic pressure causing

inactivation of protein [67]. Despite the seemingly not different expression rate between epsilon and zeta protein, the environment pressure caused by the presence of a plant extract might reduce the functional epsilon protein much lower than the expression rate reduction. Inactivation of antitoxin epsilon will let zeta toxin protein becomes its toxic form that will have an effect on cell wall biosynthesis.

Cell wall biosynthesis is initiated with a precursor UDP-N-acetylglucosamine (UNAG) that will be catalyzed by *MurA* enzyme to produce UDP-acetylglucosamine enol pyruvate (EP-U-NAG) by enol pyruvate addition at C3'-OH. An UDP-muramic acid (UNAM) is synthesized by catalyzing EP-UNAG with *MurB* enzyme. The UNAM is an active sugar providing a site for peptide chain attachment on glycan resulting peptidoglycan structure. However, the presence of zeta toxin will induce phosphorylation of UNAG at C3'-OH causing inability of *MurA* to produce EP-UNAG by catalyzing UNAG [68, 69]. This intervention will disrupt biosynthesis of cell wall resulting in incomplete or disorder structure of bacterial cell wall observed in this study. In addition, increasing of autolysin protein (encoded by *lytA* gene) as Murein hydrolase protein [56] will increase disruption of bacterial cell wall by disrupting peptide-glycan bonds causing cell wall degradation. Combination of these pathways might determine degrees of cell wall damage in which more damage was observed on ATCC 49619 having epsilon-zeta system and autolysin than MDRSP 2506 which did not have an epsilon-zeta system activity.

Another ultrastructure change found in this study was long-chain formation as the commonest morphology of MDRSP 2506. Previous studies have reported that autolysin B protein encoded by *lytB* gene was responsible in separation of daughter cells during cell division of *S. pneumoniae* resulting in frequent diplococcus morphology. Decreasing of *lytB* expression resulted in a chain morphology of *S. pneumoniae*, from a diplococcus to a more likely streptococcus (long-chain formation) [70]. Cell membrane folding observed in this study was in concordance with a previous study that reported the decrease in cell membrane integrity as a mechanism of action of antimicrobial agents. Some antimicrobial agents were reported to reduce integrity and permeability of the cell membrane causing cell membrane folding and released from cell wall [71]. In addition, long-chain structure was reported previously to reduce virulence in *Enterococcus faecalis* by increasing its susceptibility to phagocytosis and no longer causing lethality in a zebrafish model. Moreover, long-chain morphology will reduce the chance of bacterial dispersion due to phagocytic susceptibility [72]. Long-chain formation observed in MDRSP 2506 after *L. inermis* ethyl acetate treatment might also reduce virulence by being more susceptible to phagocytosis compared to diplococcus formation.

According to LC MS/MS results, this study found that phenolic compound was the commonest phytochemical compound in *L. inermis* ethyl acetate extract. As comparison, ethanol extract of *L. inermis* was also processed for LC MS/MS (S6 Fig and S4 Table) resulting in similar results with ethyl acetate extract, where phenolic compound was the most prevalent phytochemical compound observed. Among the phytochemical compounds found in this study, 2-Hydroxy-1,4-naphthoquinone (Lawsone), observed in ethanol extract, was a compound reported to have antimicrobial activity against *Bacillus cereus*, *Listeria monocytogenes*, *Salmonella enterica*, *Shigella sonnei*, *Staphylococcus epidermidis*, dan *S. intermedius* [73]. In spite of Lawsone being reported of having antimicrobial activity in previous study, this compound showed weak antibacterial activity against *S. pneumoniae* as described in this study. Other phytochemical compounds, such as derivate of benzene, napthoquinone, toluene, benzoate acid, saponins, flavonoids, and steroids in *L. inermis* extract [14] may give stronger antibacterial activity against *S. pneumoniae*. As reported previously, derivate of naphthalene and benzene showed antimicrobial and anti-inflammatory activity [74]. Moreover, antagonistic and synergistic activity might need to be assessed among phytochemical compounds in *L. inermis*

extract. Previous studies have reported that among phytochemical compounds in plant extract can produce synergistic and antagonistic of antimicrobial activity [21].

## Conclusion

In conclusion, this study discovered that plant extract with ethanol and ethyl acetate solvent had more frequent antibacterial activity compared to n-hexane and aqueous. Furthermore, *Lawsonia inermis* ethyl acetate extract had the strongest antibacterial activity showing an MIC value of 0.16 mg/ml with bactericidal activity observed at 0.16 mg/ml at 1-hour incubation. Our results described the mechanisms of action of *L. inermis* ethyl acetate extracts including bacterial lysis, antibiofilm formation, increasing of autolysin protein encoded by *lytA* gene, and decreasing of toxin-antitoxin zeta-epsilon expression encoded by *peZT* and *peZA*. Moreover, ultrastructure changes including morphological changes, decreasing of cell membrane integrity, and cell wall disruption can eventually lead to cell apoptosis and cell death.

## Supporting information

**S1 Fig. Inhibition zones of plant extracts on *S. pneumoniae* ATCC 49619.**
(DOCX)

**S2 Fig. Ultrastructure changes on MDRSP 2506 caused by *L. inermis* ethyl acetate extract.**
(DOCX)

**S3 Fig. Ultrastructure changes on ATCC 49619 caused by *L. inermis* ethyl acetate extract.**
(DOCX)

**S4 Fig. Ultrastructure changes of cell membrane on ATCC 49619 caused by *L. inermis* ethyl acetate.**
(DOCX)

**S5 Fig. Ultrastructure changes on MDRSP caused by *L. inermis* ethanol extract.**
(DOCX)

**S6 Fig. Chromatogram of *L. inermis* ethanol extract obtained with LC MS/MS.**
(DOCX)

**S1 Table. MIC and MBC value of plant extracts showed MIC value >1mg/ml on ATCC 49619 tested on MDRSP strain.**
(XLSX)

**S2 Table. Phytochemical compound of ethyl acetate *L. inermis* extract with confident level of chemical formula more than 75%.**
(XLSX)

**S3 Table. Phytochemical compound of ethyl acetate *L. inermis* extract with confident level of chemical formula less than 75%.**
(XLSX)

**S4 Table. Phytochemical compound of ethanol *L. inermis* Linn. extract.**
(XLSX)

**S1 Dataset.**
(XLSX)

## Acknowledgments

We thank to Rohayati, Smitha Mirsyad Warsadiharja, and Pak Azhar for laboratory assistance in plant extraction and laboratory testing. We also gratefully thank to Ade Irwan Rifai, Sodri, Muhammad Dailami, and Diyan for helping this study in specimen collection.

## Author Contributions

**Conceptualization:** Wisnu Tafroji, Dodi Safari.

**Data curation:** Wisnu Tafroji, Nur Ita Margyaningsih, Miftahuddin Majid Khoeri, Korrie Salsabila.

**Formal analysis:** Wisnu Tafroji, Dodi Safari.

**Funding acquisition:** Wisnu Tafroji, Dodi Safari.

**Investigation:** Amin Soebandrio, Dodi Safari.

**Methodology:** Wisnu Tafroji, Wisiva Tofriska Paramaiswari, Yayah Winarti, Hanifah Fajri Maharani Putri.

**Supervision:** Nurjati Chairani Siregar, Amin Soebandrio, Dodi Safari.

**Writing – original draft:** Wisnu Tafroji.

**Writing – review & editing:** Wisnu Tafroji, Dodi Safari.

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
