## [Decision Letter · Decision Letter 0]

23 Jun 2022

PONE-D-22-12448Antibacterial activity of medicinal plants in Indonesia on Streptococcus pneumoniaePLOS ONE

Dear Dr. Tafroji,

Thank you for submitting your manuscript to PLOS ONE. After careful consideration, we feel that it has merit but does not fully meet PLOS ONE’s publication criteria as it currently stands. Therefore, we invite you to submit a revised version of the manuscript that addresses the points raised during the review process. I would suggest you have your manuscript undergo language editing, as pointed out by reviewers. Considering the conflicting reviews, the authors are advised to provide a stronger justification of their study, by formulating a clearer hypothesis.

We look forward to receiving your revised manuscript.

Kind regards,

Olivier Habimana

Academic Editor

PLOS ONE

Journal Requirements:

"This study was funded by National Geographic Society through Young Explorer Program Grant number: HJ-081ER-17, Indonesia Toray Science Foundation (ITSF) through Science and Technology Research Grant 2018, and National Research and Innovation Agency Republic of Indonesia. We thank Rohayati, Smitha Mirsyad Warsadiharja, and Pak Azhar for laboratory assistance in plant extraction and laboratory testing. We also gratefully thank to Ade Irwan Rifai, Sodri, Muhammad Dailami, and Diyan for helping this study in specimen collection."

"WT received funding from National Geographic Society through The Young Explorer Grants; Grant no. HJ-081ER-17 (https://www.nationalgeographic.org/) and Indonesia Toray Science Foundation (ITSF) through Science and Technology Research Grant 2018 (https://itsf.or.id/winners).

DS received funding from National Research and Innovation Agency Republic of Indonesia (https://international.brin.go.id/) 

6. Please remove your figures from within your manuscript file, leaving only the individual TIFF/EPS image files, uploaded separately.These will be automatically included in the reviewers’ PDF.

7. Please upload a copy of Supporting Information Table S2, S3 and S4 which you refer to in your text on page 15. 

Reviewers' comments:

Reviewer's Responses to Questions

**Comments to the Author**

1. Is the manuscript technically sound, and do the data support the conclusions?

Reviewer #1: Yes

Reviewer #2: Partly

Reviewer #3: Partly

2. Has the statistical analysis been performed appropriately and rigorously? 

Reviewer #1: I Don't Know

Reviewer #2: I Don't Know

Reviewer #3: Yes

3. Have the authors made all data underlying the findings in their manuscript fully available?

Reviewer #1: Yes

Reviewer #2: Yes

Reviewer #3: Yes

4. Is the manuscript presented in an intelligible fashion and written in standard English?

Reviewer #1: No

Reviewer #2: Yes

Reviewer #3: Yes

5. Review Comments to the Author

Reviewer #1: There is a lack of novelty in the manuscript. Why authors did this study with plant extract? what are the photochemical compounds in these extracts? many articles published regarding this topic and what is important in this study?

I recommend the rejection of this article.

Reviewer #2: In the manuscript entitled " Antibacterial activity of medicinal p 1 lants in Indonesia on Streptococcus pneumoniae", the authors prepared extracts from many different types of medical plants using different solvents and tested the antibacterial activity of these extracts against Streptococcus pneumoniae. Most conclusions are supported by appropriate results. I recommend that the manuscript can be considered to accept for publication after the following issues have been addressed:

1. There are many grammar errors in the manuscript.

2. Using commas as the decimal point is not an international way. For example, MIC 0,16 mg/mL is suggested to be modified as “0.16 mg/mL”. Also, the number in line 65, 400.000 is suggested to be modified as “400,000” or “400000”.

3. In line 66, the authors wrote “Drug resistant Streptococcus pneumoniae was reported in Indonesia”, and in line 67, the authors wrote “many drug resistant Streptococcus pneumoniae was reported in Indonesia”. This sentence is repeated.

4. In line 72, such as what bacteria?

5. In line 79, the word “were” should be deleted.

6. In the method section, line 110, the extraction was done using hexane, and followed by acetate, ethanol…, or the extraction was done in hexane, acetate, ethanol…., respectively to get different extractions?

7. In the method section, line 132, why did the authors choose “30 µg” vancomycin as a positive control for the disc diffusion? The authors should explain this. Otherwise, it is hard to judge if 30 µg vancomycin is a suitable control. Also, in the later tests, the authors adopted 4 µg/mL vancomycin as control. Similarly, the authors should explain why they choose “4 µg” vancomycin as a control? Why do you think it is a suitable control?

8. This work used DMSO as the solvent to dissolve plant extracts and used them to do antibacterial tests. The application of these solutions might be a problem DMSO is toxic to cells. It is suggested to use safer solvents if possible.

9. In Fig. 1A, which plant extracts were used to get these data? It needs to be clarified.

10. In Fig. 3, where are the curves of 1 X MIC and 2X MIC? If these curves are overlapped with that of 10 X MIC, please describe it in the main text. If 1 X MIC and 2X are different to that of 10 X MIC, please add the data.

11. In the Mechanism of Actions section, Bacterial lysis, the authors used the growth curves of bacteria by measuring absorbance at 595 nm, and saying bacterial lysis happened because of the decrease of the absorbance at 595 nm. Why? I do not agree with this, because it is a normal growth curves measurement. The decrease of the absorbance only shows the inhibitive effects of the drug you use, but the mechanism behind is unknown. This part needs serious revision.

12. In Fig.5, the description of a to e should be added in the figure caption.

Reviewer #3: The finding is interesting, however, there are some minor mistakes have been observed (like grammar, spellings, commas, pronunciations etc.). The authors are requested to kindly correct all those errors. Also, there is a need to update the references. over-all the manuscript looks good. Tables and graphs are fine and beautifully presented. I recommend that this manuscript can be accepted for publication after minor revisions. Congratulations to authors for this wonderful work.

6. PLOS authors have the option to publish the peer review history of their article (what does this mean?). If published, this will include your full peer review and any attached files.

Reviewer #1: No

Reviewer #2: No

Reviewer #3: No

---

## [Author Response · Author response to Decision Letter 0]

8 Aug 2022

Thank you for the support comments and suggestion given by editors and reviewers to improve overall quality of this manuscript. We appreciate all the feedbacks and we hope the revision we made will be sufficient to make our manuscript acceptable for publication in the PLOS ONE

---

## [Decision Letter · Decision Letter 1]

18 Aug 2022

PONE-D-22-12448R1Antibacterial activity of medicinal plants in Indonesia on Streptococcus pneumoniaePLOS ONE

Dear Dr. Tafroji,

Thank you for submitting your manuscript to PLOS ONE. After careful consideration, we feel that it has merit but does not fully meet PLOS ONE’s publication criteria as it currently stands. Therefore, we invite you to submit a revised version of the manuscript that addresses the points raised during the review process.

**It should be pointed out that it is expected that the authors address all comments made by all reviewers on a point-by-point basis. To the author's benefit, it would be to address the comments raised by reviewer 1 concerning the justification of the study's experimental design and purpose. I highly recommend that the authors present a complete rebuttal addressing all remarks before a decision can be made.**

We look forward to receiving your revised manuscript.

Kind regards,

Olivier Habimana

Academic Editor

PLOS ONE

Reviewers' comments:

Reviewer's Responses to Questions

**Comments to the Author**

1. If the authors have adequately addressed your comments raised in a previous round of review and you feel that this manuscript is now acceptable for publication, you may indicate that here to bypass the “Comments to the Author” section, enter your conflict of interest statement in the “Confidential to Editor” section, and submit your "Accept" recommendation.

Reviewer #1: (No Response)

Reviewer #3: (No Response)

2. Is the manuscript technically sound, and do the data support the conclusions?

Reviewer #1: (No Response)

Reviewer #3: (No Response)

3. Has the statistical analysis been performed appropriately and rigorously? 

Reviewer #1: (No Response)

Reviewer #3: (No Response)

4. Have the authors made all data underlying the findings in their manuscript fully available?

Reviewer #1: (No Response)

Reviewer #3: (No Response)

5. Is the manuscript presented in an intelligible fashion and written in standard English?

Reviewer #1: (No Response)

Reviewer #3: (No Response)

6. Review Comments to the Author

Reviewer #1: There is a lack of novelty in the manuscript. Why authors did this study with plant extract? what are the photochemical compounds in these extracts? many articles published regarding this topic and what is important in this study?

I recommend the rejection of this article.

Reviewer #3: This manuscript can be accepted in its present form. The authors have carefully revised the manuscript and responded all the queries

7. PLOS authors have the option to publish the peer review history of their article (what does this mean?). If published, this will include your full peer review and any attached files.

Reviewer #1: **Yes: **#########

Reviewer #3: No

---

## [Author Response · Author response to Decision Letter 1]

23 Aug 2022

I have revised the response to reviewer as the file's name: Revised response to reviewers.docx. We apologise to reviewer #1 for not responding the question completely. We hope the revised response will help the decision of this manuscript.

We thank to all reviewers and editors for the comments and suggestions addressed to increase quality this manuscript. we hope the revision we made will be sufficient to get this manuscript published in PLOSONE.

---

## [Editor Report · Decision Letter 2]

24 Aug 2022

Antibacterial activity of medicinal plants in Indonesia on Streptococcus pneumoniae

PONE-D-22-12448R2

Dear Dr. Tafroji,

We’re pleased to inform you that your manuscript has been judged scientifically suitable for publication and will be formally accepted for publication once it meets all outstanding technical requirements.

Kind regards,

Olivier Habimana

Academic Editor

PLOS ONE

Additional Editor Comments (optional):

The response to all reviewers, as well as the revisions to original manuscript satisfactory.

---

## [Editor Report · Acceptance letter]

1 Sep 2022

PONE-D-22-12448R2 

Antibacterial activity of medicinal plants in Indonesia on *Streptococcus pneumoniae*

Dear Dr. Tafroji:

I'm pleased to inform you that your manuscript has been deemed suitable for publication in PLOS ONE. Congratulations! Your manuscript is now with our production department. 

Kind regards, 

on behalf of

Dr. Olivier Habimana 

Academic Editor

PLOS ONE